# Quantifying spatial uncertainty to improve soil predictions in data-sparse regions

Kerstin Rau<sup>1,3,4</sup>, Katharina Eggensperger<sup>2,3</sup>, Frank Schneider<sup>2,4</sup>, Michael Blaschek<sup>5</sup>, Philipp Hennig<sup>2,3,4</sup>, and Thomas Scholten<sup>1,3</sup>

Albertstraße 5, Freiburg im Breisgau, 79104, Baden-Württemberg, Germany

Correspondence: Kerstin Rau (kerstin.rau@uni-tuebingen.de)

**Abstract.** Artificial Neural Networks (ANNs) are valuable tools for predicting soil properties using large datasets. However, a common challenge in soil sciences is the uneven distribution of soil samples, which often results from past sampling projects that heavily sample certain areas while leaving similar yet geographically distant regions under-sampled. One potential solution to this problem is to transfer an already trained model to other similar regions. Robust spatial uncertainty quantification is crucial for this purpose, yet often overlooked in current research. We address this issue by using a Bayesian deep learning technique, Laplace Approximations, to quantify spatial uncertainty. This produces a probability measure encoding where the model's prediction is deemed reliable, and where a lack of data should lead to a high uncertainty. We train such an ANN on a soil landscape dataset from a specific region in southern Germany and then transfer the trained model to another unseen but to some extend similar region, without any further model training. The model effectively generalized alluvial patterns, demonstrating its ability to recognize repetitive features of river systems. However, the model showed a tendency to favor overrepresented soil units, underscoring the importance of balancing training datasets to reduce overconfidence in dominant classes. Quantifying uncertainty in this way allows stakeholders to better identify regions and settings in need of further data collection, enhancing decision-making and prioritizing efforts in data collection. Our approach is computationally lightweight and can be added post-hoc to existing deep learning solutions for soil prediction, thus offering a practical tool to improve soil property predictions in under-sampled areas, as well as optimizing future sampling strategies, ensuring resources are allocated efficiently for maximum data coverage and accuracy.

# 1 Introduction

Machine learning (ML) has become an indispensable tool in scientific research, leading to significant advances in many fields, including soil science (Zhang et al., 2022). Since the early 2000s, ML methods have been steadily integrated into soil mapping

<sup>&</sup>lt;sup>1</sup>Department of Geoscience, University of Tübingen, Rümelinstraße 19-23, Tübingen, 72070, Baden-Württemberg, Germany <sup>2</sup>Department of Computer Science, University of Tübingen, Maria-von-Linden-Straße 6, Tübingen, 72076, Baden-Württemberg, Germany

<sup>&</sup>lt;sup>3</sup>Cluster of Excellence Machine Learning: New Perspectives for Sciene, University of Tübingen, Maria-von-Linden-Straße 6, Tübingen, 72076, Baden-Württemberg, Germany

<sup>&</sup>lt;sup>4</sup>Tübingen AI Center, University of Tübingen, Maria-von-Linden-Straße 6, Tübingen, 72076, Baden-Württemberg, Germany

<sup>&</sup>lt;sup>5</sup>Department 9: State Authority for Geology, Mineral Resources and Mining (LGRB), Regional Council Freiburg,

20 (McBratney et al., 2003; Scull et al., 2003; Behrens et al., 2005). Over the past two decades, the use of ML in soil science has grown substantially, reflecting its increasing importance and effectiveness (Minasny and McBratney, 2016; Rentschler et al., 2022; Zhang et al., 2022; Kebonye et al., 2023; Taghizadeh-Mehrjardi et al., 2024). However, new research challenges have emerged as ML methods become more widely used. One key challenge is improving model interpretability in order to promote scientific knowledge (Padarian et al., 2020b). Chen et al. (2022) further added that future research should focus on making the most of legacy datasets, using smarter sampling strategies, improving model accuracy and interpretability, and developing advanced mapping methods to create detailed and high-quality soil maps. In line with this, Bohn and Miller (2024) showed that locally enhanced, bottom-up oriented DSM approach has been shown to deliver higher accuracy compared to both conventional soil maps and global DSM products in many cases.

Using legacy datasets effectively means applying data from already sampled areas to predict conditions in similar but unsampled regions. This extrapolation process has been discussed for many years. For example, Lagacherie et al. (1995) pointed out the need to develop self-learning systems that dynamically adapt predictions. Bui and Moran (2003) demonstrated that existing soil maps can be combined with environmental and geological data to extend their usefulness beyond their original boundaries. Additionally, Scull et al. (2005) showed with classification trees that this technique allows soil experts to focus field mapping on unique areas and efficiently extrapolate soil-landscape relationships, making it a valuable tool for soil surveys. Extrapolation approaches also address the challenges of traditional soil mapping, which relies on cartographers manually surveying landscapes, a process that is both costly and time-consuming. These methods offer a particularly cost-effective solution for predicting soil classes in regions with limited data, helping to fill the gaps in soil maps and improving the efficiency of digital soil mapping (DSM) (Taghizadeh-Mehrjardi et al., 2022). For example, decision trees demonstrated a 46.00 % overall accuracy for extrapolating soil subgroups using digital mapping methods, making them a cost-effective option for areas with limited data or challenging sampling conditions (Neyestani et al., 2021). Similarly, multinomial logistic regression and classification trees have been used successfully to extrapolate soil classes (Abbaszadeh Afshar et al., 2018; Lemercier et al., 2012; Grinand et al., 2008). To summarize, the increasing adoption of machine learning is driven not only by its relevance to soil science but also by its ability to significantly reduce the effort required for mapping, especially in large or hard-to-access areas (Hewitt, 1993; Grunwald et al., 2011; Stumpf et al., 2017).

More advanced methods, in particular Artificial Neural Networks (ANNs), have proven effective for extrapolation in soil mapping. For instance, the study by de Arruda et al. (2016) demonstrated the potential of ANNs to produce digital soil maps, providing initial classifications for unexplored areas. Building on this, Coelho et al. (2021) introduced an innovative methodology that combined georeferenced soil profile point data and ANN models for extrapolation tasks. Responding to the growing demand for high-resolution soil maps in, for instance, precision agriculture, environmental management, and land-use planning, ANNs are becoming more and more popular due to their ability to process large amounts of data and provide predictions comparably fast (Haykin, 1994; Schmidhuber, 2015; Silveira et al., 2013). Brungard et al. (2015) demonstrated the superior accuracy of complex models containing neural networks in predicting soil taxonomy classes compared to simpler models. Similarly, Zhu (2000) highlighted the capability of ANNs in generating high-resolution soil maps.

Despite their advantages, one notable challenge is the lack of inherent interpretability (Heung et al., 2016). As "black box" models. ANNs make predictions through complex internal processes that are difficult to understand and interpret. Recent studies have addressed this limitation by introducing model-agnostic interpretation techniques and game theory-based Shapley additive explanations (SHAP), which provide valuable insights into the relationships between environmental covariates and model predictions (Padarian et al., 2020a; Wadoux and Molnar, 2022). In addition, ANNs typically lack built-in uncertainty quantification, which complicates the evaluation of their predictive reliability and may lead to misinterpretations or suboptimal decision-making (Guo et al., 2017). They often produce overly confident predictions, sometimes reaching 100.00 % certainty, even when the input data is flawed or noisy (Breiman, 2001; Nguyen et al., 2015; Hein et al., 2019). In the context of DSM, this issue is compounded by the broader challenge of quantifying spatial uncertainty in soil maps (Hengl et al., 2017; Wadoux et al., 2020; Rau et al., 2024). Between 2017 and 2022, only 35.00 % of studies that addressed significant DSM tasks incorporated uncertainty in their analysis (Belkadi and Drias, 2023). Similarly, while DSM research is expanding in countries such as India and Iran, the integration of uncertainty mapping remains limited. In India, only 34.00 % of DSM studies include uncertainty maps, while in Iran, fewer than 20.00 \% address uncertainty (Zeraatpisheh et al., 2020; Dash et al., 2022). Typically, these maps then present just an overall accuracy expressed as a single statistical measure, often derived through cross-validation techniques, an iterative process that partitions the training data into multiple subsets to repeatedly train and validate the model to estimate overall performance uncertainty (Wadoux et al., 2020). Although this approach offers some insight, it falls short, especially for applications involving unbalanced datasets. This gap has led to calls for more detailed uncertainty analysis (Meyer and Pebesma, 2022), particularly for tasks involving extrapolation to new areas because of poor uncertainty performance in such contexts (Grinand et al., 2008). Some recent studies have made strides in incorporating uncertainty quantification. For instance, Carvalho Monteiro et al. (2023) and van der Westhuizen et al. (2023) have demonstrated progress in quantifying uncertainties for Random Forests, while Saygın et al. (2023) have explored the use of ANNs. However, many of these methods rely on variance estimates, which fail to adequately address critical issues such as model overconfidence. This problem has emerged in the study by Schmidinger and Heuvelink (2023) that ANNs produce overly optimistic probabilistic predictions, resulting in low-reliability scores. Additionally, these approaches frequently neglect spatial uncertainty, an essential aspect of practical soil mapping (Bao et al., 2024). The most commonly used methods for uncertainty quantification in DL algorithms, particularly in ANNs, include Monte Carlo (MC) Dropout, ensemble methods, and full Bayesian approaches. These methods, while effective, often require significant computational resources and memory (Abdar et al., 2021). These techniques have begun to gain traction in soil science applications, particularly for estimating uncertainty in soil moisture retrieval or soil spectral models (Li et al., 2023). For example, Padarian et al. (2022) and Huang et al. (2025) utilized these approaches to assess uncertainty in their models, demonstrating their relevance and utility despite the computational demands. These findings underscore the urgent need for methodological advancements that go beyond variance estimation to also tackle overconfidence together with spatial uncertainty while remaining computationally efficient and easy to integrate into existing workflows. Such improvements are crucial to ensure that machine learning models for DSM provide both accurate and reliable predictions. Our previous work Rau et al. (2024) introduced for DSM the Last-Layer Laplace Approximation (LLLA), a computationally efficient technique that addresses these challenges. Building on this methodological foundation, the current study applies an ANN model to an

extrapolation task, predicting soil units non-adjacent target area outside the training area. To identify and correct the overconfidence of the ANN and perform a spatial analysis of the model's predictions and associated uncertainties, we use the LLLA, providing corrected uncertainty estimates for every pixel in the target area. Through this, we assess the transferability of the ANN by improving its interpretability and reliability for soil mapping tasks. Ultimately, our work aims to promote more robust, accurate, and insightful applications of DSM.

## 2 Material and Methods

## 2.1 Study area

100

110

This study investigates two regions in central Baden-Württemberg, Germany, near the city of Tübingen. The reference area is located northwest of the city, and the target area lies to the southwest, as shown in Figure 1 (A). These regions were chosen because they share similar geology, climate, and cultural development, making them suitable for comparative analysis. The reference area, named after the Goldersbach stream, covers 8.86 km<sup>2</sup> with an average elevation of 445.51 m above sea level, ranging from 325.31 m to 552.48 m. It represents the lower section of the upstream part of the Goldersbach River and its catchment. The main land use since the 19th century in this area is forestry and since 1972 it has been part of a nature park. The target area, named after the Bühlertalbach stream, is larger, covering  $18.5\,\mathrm{km}^2$  with an average elevation of  $498.26\,\mathrm{m}$ above sea level, ranging from 388.86 m to 583.04 m. It includes the entire Bühlertalbach stream valley, from its upstream to downstream sections. Similar to the reference area, forestry is the main land use, and this area is extensively used for forestrelated activities. Both areas have the same underlying geology, belonging to the Middle and Upper Keuper series, which consist of layers of sandstone, claystone, and marlstone, creating typical soil patterns of the Keuperbergland. The climate in both regions is cool temperate moist, with an average annual temperature between 8.3 °C and 8.7 °C and annual precipitation ranging from 740 mm to 770 mm. The target area was deliberately chosen to be larger, encompassing the entire catchment of the Bühlertalbach River. This strategic decision allows for the investigation of how predictions and findings extend beyond the upstream areas on which the reference area is based. By including the full catchment, this approach provides a broader and more comprehensive understanding of processes in similar but not equal landscapes.

# 2.2 Data

Figure 1 illustrates the distribution of soil units across the reference area (blue, subfigure (B)) and target area (red, subfigure (C)), with each number corresponding to a specific soil unit and its associated characterization. The classification of the soil types in these units follows the LGRB soil classification system, a local variant of the German soil classification KA5, which is structured around soil formation processes and properties (Eckelmann et al., 2005). Within the reference area, there are eight distinct soil units, alongside an urban zone represented as unit 0. In contrast, the target area exhibits greater diversity, comprising 14 unique soil units. A comprehensive description of all these units is provided in Table 1, including their correspondence with the World Reference Base (WRB) soil classification system (IUSS Working Group, (2022).

Figure 1. (A) Digital elevation model of the study area with the location of the study area in Germany, reference area in blue, target area in red, (B) and (C) Soil unit maps over the reference and target areas, created by the *State Authority for Geology, Mineral Resources and Mining (LGRB) Baden-Württemberg*.

For international understanding, we use the WRB classification system. The soil unit maps, initially sourced from the *State Authority for Geology, Mineral Resources and Mining (LGRB)*, were provided in vector format. To facilitate our analysis, these polygons were converted into raster files using a rasterization process based on digital elevation grids. The original map scale of 1:50,000 was rasterized to produce a  $10 \, \text{m} \times 10 \, \text{m}$  resolution. It should be noted that this study is based entirely on pixel-based soil unit prediction using these rasterized soil maps as training and validation labels rather than direct field observations. To enhance the performance of the neural network and ensure detailed analysis, spatially dense covariate data were required for the entire region. For this purpose, digital elevation models (Figure 1 (A)) were used for both areas. These models offer a  $10 \, \text{m}$  resolution and serve as the basis for calculating topographic indices, also at  $10 \, \text{m}$  resolution. The variable selection was informed by local expert geographical knowledge, guided by commonly used proxies representing the SCORPAN model introduced by McBratney et al. (2003), which draws upon Jenny (1941). In addition, we included spectral indices based on satellite data from the *Copernicus Sentinel-2* program. Since 2017, *Sentinel-2* provides data in 13 spectral bands with a 5-day revisit time. For this study, we focused on the visible (R, G, B) and near-infrared bands, which have a  $10 \, \text{m}$  resolution. Using

these bands, we calculated indices such as the Normalized Difference Vegetation Index (NDVI) to measure vegetation cover. To ensure robust data representation and reduce the impact of outliers, we computed the median values of these indices for the time series of cloud-free images from March to May 2019. We also included geological maps, scaled at 1:50,000, provided by the *LGRB*. These maps were rasterized in the same way as the soil unit maps. Table 2 summarizes the indices and variables used for the ANN as covariates and their respective references. To compare the covariates in the reference and target areas, we applied the cosine similarity index, as outlined by Schütze et al. (2008). This method, which measures similarity on a scale from -1 (completely opposite) to 1 (identical), resulted in a mean value of 0.85. Using this score confirmed a strong similarity between the areas. In addition, we collaborated with experts from the *LGRB*, whose extensive regional knowledge ensured the appropriate selection of study areas. Both the similarity assessment and the expert consultation were carried out in recognition of the fact that, even at the local scale, it is crucial to apply models only where they are valid, a principle already established in global-scale research (Ludwig et al., 2023).

# 2.3 Model design

135

140

160

Artificial Neural Networks (ANNs) originated in the field of image recognition, particularly for classification tasks (Goodfellow et al., 2016). These models are highly effective at identifying patterns and relationships in data, even without prior domain knowledge, and excel in handling large datasets. ANNs are composed of layers of neurons leveraging activation functions to learn complex patterns. The structure of an ANN can vary widely in terms of its architecture, the number and type of layers, their dimensions, and the activation functions used. Since our research prioritizes understanding uncertainty in machine learning models applied to soil data rather than optimizing model performance, we opted for a straightforward design: a fully connected multilayer perceptron, as described in Table 3. For the hidden layers, we employed the rectified linear unit (ReLU) activation function, which is defined as:

$$ReLU(x) = max(0, x)$$

where x is the input to a neuron (Fukushima, 1969; Glorot et al., 2011; Nair and Hinton, 2010). For training and validation, we used data from the reference area, which includes 33 covariates (listed in Table 2) and soil unit labels. The reference area comprises eight distinct soil units, which the model aims to predict. To evaluate the model's performance, we tested it on the ground truth map of soil units from the target area (Figure 1 (C)). The training and validation dataset consisted of 142,569 data points, i.e. the number of raster cells, which was separated through random sampling to a 70%-30% split, while the test dataset contained 378,214 data points. We used the architecture mentioned above. A detailed description of the model tuning protocol is provided in Rau et al. (2024), where the method was first tested in a simplified, controlled soil classification setup. The optimized hyperparameters derived from this process were successfully transferred and applied to the reference area, yielding excellent results. To enhance the model's robustness and prevent overfitting, we implemented an early stopping criterion. The training process was halted when the model's training accuracy, defined as the percentage of correctly predicted pixels, exceeded 95.00%, and no significant improvement in test dataset accuracy was observed.

| Class | Label | German Soil Classification                                 | WRB-Classification | Detailed information                                                      |
|-------|-------|------------------------------------------------------------|--------------------|---------------------------------------------------------------------------|
| no.   |       |                                                            |                    |                                                                           |
| 0     | None  | None                                                       | None               | Ablation, order, settlement                                               |
| 1     | A1    | Brauner Auenboden, Auenbraunerde                           | Fluvisol, Cambisol | partly with gleying in the near subsoil, of al-                           |
|       |       |                                                            |                    | luvial sand and alluvial loam                                             |
| 2     | A3    | Auengley, Auenpseudogley-Auengley, Brauner                 | Fluvisol           | from alluvial sand and alluvial clay                                      |
|       |       | Auenboden-Auengley                                         |                    |                                                                           |
| 3     | A7    | Auenbraunerde, Auenparabraunerde                           | Cambisol           | from older alluvial sediment                                              |
| 4     | B2    | Braunerde, Pelosol-Braunerde, Pseudogley-                  | Cambisol           | from solifluction soils, partly alluvial and                              |
|       |       | Braunerde                                                  |                    | flood loam                                                                |
| 5     | B4    | Braunerde, Podsol-Braunerde                                | Arenosol           | mostly podzolic, from sandstone, debris-rich                              |
|       |       |                                                            |                    | fluvial soils and slope debris                                            |
| 6     | D1    | Pelosol, Braunerde-Pelosol, Pseudogley-                    | Luvisol-Vertisol   | from solifluction soils, subordinate from al-                             |
|       |       | Pelosol                                                    |                    | luvial debris                                                             |
| 7     | K1    | Kolluvium                                                  | Anthrosol          | partly over Braunerde and Parabraunerde,                                  |
|       |       |                                                            |                    | from alluvial deposits over solifluction soils                            |
| 8     | K2    | Pseudogley-Kolluvium, Gley-Kolluvium                       | Gleyic Anthrosol   | from alluvial deposits                                                    |
| 9     | L2    | Parabraunerde, Braunerde-Parabraunerde,                    | Luvisol            | of loess loam and loess-loam-rich solifluc-                               |
|       |       | Pseudogley-Parabraunerde                                   |                    | tion soils                                                                |
| 10    | L3    | Parabraunerde, Pelosol-Parabraunerde,                      | Luvisol            | from solifluction soils and slope debris                                  |
|       |       | Terra fusca-Parabraunerde, Pseudogley-                     |                    |                                                                           |
| 1.1   | NT1   | Parabraunerde                                              |                    | 6                                                                         |
| 11    | N1    | Ranker und Braunerde-Ranker                                | Leptosol-Cambisol  | from sandstone                                                            |
| 12    | S1    | Pseudogley, Braunerde-Pseudogley, Pelosol-                 | Planosol-Cambisol  | from solifluction soils, partly Pleistocene al-                           |
| 12    | 00    | Pseudogley                                                 | DI 11 ' 1          | luvial debris                                                             |
| 13    | S2    | Pseudogley, Parabraunerde-Pseudogley                       | Planosol-Luvisol   | of loess loam and loess-loam-rich solifluc-<br>tion soils                 |
| 1.4   | 771   | D 1: D1 1D 1:                                              | T 4 137 4 1        |                                                                           |
| 14    | Z1    | Pararendzina, Pelosol-Pararendzina, Braunerde-Pararendzina | Leptosol-Vertisol  | from solifluction soils and slope debris,<br>partly from landslide masses |
| 15    | D2    |                                                            | Lantagal           |                                                                           |
| 15    | R3    | Rendzina und Terra fusca-Rendzina                          | Leptosol           | from river gravels                                                        |

Table 1. Detailed description of the soil units

| Environmental input data |                                                                                                                                                                                                                                                                                                                                                                                                            | Definition after                                                               |
|--------------------------|------------------------------------------------------------------------------------------------------------------------------------------------------------------------------------------------------------------------------------------------------------------------------------------------------------------------------------------------------------------------------------------------------------|--------------------------------------------------------------------------------|
| Topographic              | Eastness, Elevation, Northness, Slope                                                                                                                                                                                                                                                                                                                                                                      | Bauer et al. (1985)                                                            |
| indices                  | Diffuse radiation, Direct radiation, Slope discontinuities, Terrain classification index for lowlands                                                                                                                                                                                                                                                                                                      | Bock et al. (2007)                                                             |
|                          | Relative height above the depth line, Soil moisture                                                                                                                                                                                                                                                                                                                                                        | Böhner and Köthe (2003)                                                        |
|                          | Catchment area                                                                                                                                                                                                                                                                                                                                                                                             | Freeman (1991)                                                                 |
|                          | Plan curvature, Profile curvature                                                                                                                                                                                                                                                                                                                                                                          | Heerdegen and Beran (1982)                                                     |
|                          | Convergence divergence index, Crest index for lowlands, Crest index for mountain areas, Culmination line for lowlands, Culmination line for mountain areas, Elevation below the culmination line for lowlands, Elevation below the culmination line for mountain areas, Horizontal distance to the depth line, Relative hill slope position for lowlands, Relative hill slope position for mountain areas, | Köthe and Bock (2006)                                                          |
|                          | Relative altitude, Relief                                                                                                                                                                                                                                                                                                                                                                                  |                                                                                |
|                          | Depth of closed surface depressions                                                                                                                                                                                                                                                                                                                                                                        | Wang and Liu (2006)                                                            |
| Spectral indices         | Brightness index, Colouration index, Hue index, Normalized difference vegetation index, Redness index, Saturation index                                                                                                                                                                                                                                                                                    | Hounkpatin et al. (2018)                                                       |
| Geological<br>variable   | Geological map                                                                                                                                                                                                                                                                                                                                                                                             | Department 9: State Authority for Geology, Mineral Resources and Mining (LGRB) |

**Table 2.** Overview of the covariates for the neural network

| Layer        | Number of neurons | Activation function |
|--------------|-------------------|---------------------|
| Input layer  | 33                | ReLU                |
| Layer 1      | 395               | ReLU                |
| Layer 2      | 510               | ReLU                |
| Layer 3      | 489               | ReLU                |
| Output Layer | 9                 | Softmax             |

Table 3. Architecture of the Artificial Neural Network

# 2.4 Uncertainty measurement of ANNs with Last-Layer Laplace Approximation

For ANNs commonly the Softmax function in the output layer is used to convert raw scores into a probability distribution over the predicted classes. The Softmax function transforms the output of the previous layer into a vector of probabilities, essentially forming a distribution across input classes. The Softmax function is defined as follows:

$$Softmax(x_i) = \frac{\exp(x_i)}{\sum_j \exp(x_i)}$$
 (1)

where x is the vector of raw score for all classes (Bridle, 1990; Goodfellow et al., 2016). These probability values can be interpreted as uncertainty about the classification output. A higher probability indicates greater certainty, while a lower value signifies uncertainty. In other words, the ANN has predicted a class with low uncertainty and is therefore very confident about the prediction. Nevertheless, relying solely on Softmax-derived uncertainty measures has limitations, particularly in regions where the ANN encounters data points far from its training distribution, as they do not account for the uncertainty in the model's parameters or structure (Hein et al., 2019; Guo et al., 2017). To address these limitations and quantify model uncertainty, i.e. epistemic uncertainty, we employ the Last-Layer Laplace Approximation (LLLA) following Daxberger et al. (2021). This method is based on Bayesian principles and provides a computationally efficient approach to estimate posterior uncertainties for neural network parameters. The LLLA approximates the posterior distribution of the weights as

$$p(\Theta \mid D) \approx \mathcal{N}(\Theta; \Theta_{MAP}, \Sigma) \quad \text{with} \quad \Sigma := (\nabla_{\Theta}^2 \mathcal{L}(\mathcal{D}; \Theta) \mid_{\Theta_{MAP}})^{-1}.$$
 (2)

Here,  $\Theta_{MAP}$  represents the maximum a posteriori estimate of the last-layer parameters, obtained by minimizing the negative log posterior  $\mathcal{L}(\mathcal{D};\Theta)$ , typically the cross-entropy loss with an isotropic Gaussian prior. Epistemic uncertainty is captured by the LLLA through the local curvature (Hessian) of the loss: flat directions in this geometry indicate parameters that are weakly constrained by the data and thus remain uncertain. Such flatness arises in regions of limited data, structural uncertainty, or poor transferability, all of which reflect ambiguity in the posterior distribution over model parameters. The method is accessible through the open-source laplace.torch package, which facilitates easy integration into PyTorch-based workflows. The LLLA method offers key advantages: it is computationally efficient by focusing on the last layer (Kristiadi et al., 2020) and has the benefit that the point estimate ( $\Theta_{MAP}$ ) is unaffected by the uncertainty estimation, which simplifies development and tuning. We already have shown that LLLA effectively identifies areas of high uncertainty in soil classification tasks (Rau et al., 2024), making it crucial for generating uncertainty maps with uneven training data coverage.

# 3 Results & Discussion

## 190 3.1 Loss and accuracy of the ANN

In our study, we used a simple neural network architecture rather than a highly specialized one tailored to the soil classification extrapolation task. This decision reflects common scenarios where pre-built models are preferred due to their ease of use and quick deployment. Our goal was to assess and enhance the neural network's ability to extrapolate, and not to achieve the highest possible overall accuracy outperforming other state-of-the-art ANNs, an objective that could be pursued through deliberate and targeted hyperparameter optimization (Probst et al., 2019). For this reason, we focus on the spatial uncertainty at pixel level rather than the total uncertainty of the soil map, which is used in DSM (Wadoux et al., 2020). The study was therefore based on two different areas: a well-sampled reference area and a completely unsampled target area (Figure 1). This setup simulated a realistic challenge, where models are often required to make predictions in areas with limited or no prior information (Heuvelink and Webster, 2001). The results indicated that the model learned the training data effectively, achieving a low loss value of 0.01, a high training accuracy of 98.57 %, and a validation accuracy of 96.73 %. However, when applied to

the target area, the test accuracy dropped significantly to 47.38 %. This decrease was expected since there was no area-specific tuning for the neural network and our target area is substantially larger than the reference area including the entire course of the river. The reference area can thus not fully represent the target area (Warrick, 2001). Compared with other studies in DSM that use neural networks for soil classification (e.g., Zhu (2000); Behrens et al. (2005); Boruvka and Penizek (2006); Bodaghabad et al. (2015); Neyestani et al. (2021)), our model performed at an average level, consistent with our expectations due to similar parent material and climate as well as cultural development of the areas over the last centuries. These initial findings emphasize the trade-offs between simplicity and predictive performance when using simple neural networks in soil mapping applications. While these are convenient and easy to deploy, their performance is often limited in data-sparse regions. This highlights the importance of complementing neural network predictions with uncertainty quantification to effectively identify data gaps.

## 210 3.2 Prediction of the ANN

The prediction of soil units across the target area reveals several notable patterns and challenges when we compare the ANN-predicted map in Figure 2 (A) with our ground truth derived from the *LGRB* in Figure 1 (C). Not all soil units were predicted, which is expected as certain soil units (in that case soil units 1, 3, 7, 8, 10, 14, and 15) were absent from the reference area. The ANN could not predict these soil units due to its lack of training data. This phenomenon is not uncommon in practice, as soil units in complex areas often remain untested in reality (Heuvelink and Webster, 2001). For example, units 1 and 3 belong to floodplain soils, more precisely to Fluvisols after WRB, and are not included in the reference area. Similarly, soil unit 14, representing soils over the Gipskeuper formation, is also missing. The absence of certain soil units in the training dataset reveals how the neural network handles such cases and provides insight into the associated uncertainty. A detailed breakdown of the soil units is provided in Table 1.

An overall evaluation of the predictions reveals that certain regions, particularly the northern and southwestern parts of the target area, were predicted wrong (Figure 2 (B)). Interestingly, these areas correspond to the upstream and downstream sections of the river, which were areas not well represented in the reference area. We can see that predictions were more accurate in the middle stream of the river, which closely aligns with the reference area, compared to the downstream and upstream regions. This is likely due to the increasing distance from the training data. A brief and selected breakdown of the prediction of the specific soil units is now given and can be followed with the comparison of the ground truth map in Figure 1 (C) with the prediction of the ANN in Figure 2 (A) and with the confusion matrix in Figure 3.

Unit 0, which represents the settlements, was completely misclassified and often predicted as unit 2 or more often as units 4 and 6. Because unit 0 is found in the river valley and extends towards the receiving stream, these predictions are not surprising. It is interesting to see what happens to soil units 1 and 3, both alluvial soil units, neither of which occur in the reference area. Soil unit 1, a Fluvisol from alluvial sand and loam, was predicted as soil unit 2, which is the only Fluvisol in the reference area, also from alluvial sand and clay. Meanwhile, unit 3, classified as Cambisol, is often predicted as soil unit 4, representing Cambisol formed from alluvial deposits, which aligns with unit 3's characteristics as a floodplain Cambisol from older sediments. In conclusion, the model's predictions for units 1 and 3 demonstrate its ability to recognize and generalize alluvial patterns, even when specific soil units are missing from the reference data. These results suggest that floodplain soils maintain recognizable

Figure 2. (A) Prediction map of the soil unit in our target area by the ANN, (B) Comparison of the prediction with the ground truth: green means a correct prediction of the soil unit

characteristics across regions, and the model effectively learns the repetitive patterns of river systems, such as floodplains and channel deposits, during training. Unit 4 is dominant in both the reference and target areas, as confirmed by the ground truth map. Its proportion increased from 51.09 % in the ground truth from the target area to 64.21 % in the prediction by the ANN. However, this soil unit was overestimated in the central region and underestimated in the south, demonstrating the importance of spatial analysis. This is also the case for soil unit 5. This soil unit was underestimated in the central region but correctly identified in the southwest and north. Overestimations occurred in the south, and misclassifications primarily involved unit 4. This can be explained by the fact that both represent the most common soil units in Germany and the domination of 4 in the training data (Amelung et al., 2018; Wiechmann, (2000). As a result, the neural network tends to predict them more often in the output (Johnson and Khoshgoftaar, 2019). Soil unit 6 was also underestimated overall but maintained its proportional representation due to false predictions along the southern margins. The colluvial soil units 7 and 8, were not part of the training set, so they were misclassified as 4 and 6, soil unit 7 specifically as unit 6 in the southwest and unit 4 in the north. Unit 9 was extremely accurately detected in four areas in the south, east, and west but was underrepresented overall. In the southwest, areas belonging to unit 9 were often predicted as unit 13, which is also a Luvisol from loess loam.

**Figure 3.** The confusion matrix of the ANN displays true vs. predicted classifications, with diagonal values indicating correct predictions and off-diagonal values showing misclassifications.

The model effectively recognizes familiar soil units, indicating that it successfully learns and applies process-based rules of soil formation, as demonstrated by the predictions for soil units 1 and 3. However, it shows a tendency to generalize soil units based on shared properties, as seen in the misclassification of soil unit 13 as unit 9 due to their similar origin as Luvisols from loess. This suggests the model is adept at identifying broad patterns but lacks sensitivity to regional nuances and finer distinctions between similar units. The substantial regional variability in the predictions highlights the need for spatial uncertainty analyses to improve accuracy and address the model's limitations in handling less common or unfamiliar soil units.

# 3.3 Confidence of the ANN

250

Based on the previous results, especially the large distribution of correctly and incorrectly predicted classes in a unit and a non-spatial accuracy of 47.38 %, we now analyze the uncertainty of the ANN prediction of every single soil unit before applying the LLLA. In the case of an ANN, besides cross-validation methods and other techniques, a common step is to evaluate the

probability of the predicted class (Wadoux et al., 2020). This probability can be interpreted as the confidence of the model in its predictions, thus the degree of uncertainty of the model regarding the predictions per pixel (see Figure 4 (A)).

Figure 4. (A) probability of the soil unit in our target area predicted by the ANN calculated with the Softmax function, interpreted as the confidence of the ANN (B) probability after applying the LLLA, interpreted as the uncertainty of the ANN

Notably, the highest confidence values, often reaching 100 %, are observed at the borders in the south, west, and north, as well as in the central region near the river. In contrast, the intermediate regions display a more diverse confidence distribution, though the values remain generally high. This trend is reflected in the mean confidence value, which stands at 96.22 %. When examining the relationship between confidence and prediction accuracy, it can be seen that in areas where the ANN performs poorly (Figure 2 (B)), the confidence values paradoxically remain high (Figure 4 (A)). This indicates overconfidence in regions, which happens with ANNs when the training data does not represent the target area. For example, pixels where soil units are correctly predicted exhibit a mean confidence of 97.17 %, while incorrectly predicted pixels also demonstrate a high mean confidence of 95.36 %. This pattern underscores the ANN's tendency to assign high confidence to both correct and incorrect predictions, exacerbating the issue of overconfidence. Such behavior aligns with findings from previous studies, which have highlighted the tendency of ANNs to exhibit overconfidence in data-scarce regions (Kasiviswanathan et al., 2018; Hein et al., 2019; Rau et al., 2024).

Further analysis of the confidence distribution for each soil unit is presented using a violin plot in Figure 5. The blue curves represent the distribution of confidence values for each unit, focusing only on soil units present in the reference area, as these

are the only ones the ANN can predict. The width of the plot indicates where confidence values are more frequent, and the shape shows the range of these values. For most soil units, there are sharp peaks around 100 %, which means that the ANN is overly confident in all units. However, soil units 11 and 12 stand out, as the ANN also shows high confidence here, but the shape is in a wider range. This analysis highlights the issue of overconfidence of ANNs: here the ANN is too confident, even in areas where it performs poorly. This overconfidence is especially apparent in regions far from the training data or underrepresented areas. To improve the ANN's reliability, its ability to estimate its uncertainty needs to be enhanced. Further detailed analysis by soil units will be provided when comparing the ANN's predictions with those after applying the LLLA method as described in Section 2.4.

Figure 5. Distributions of the probability of predictions across the soil units (Blue: before LLLA; orange: after LLLA).

## 3.4 Uncertainty of the ANN from Last-Layer Laplace Approximation

As discussed in Section 2.4, the application of the LLLA method enabled us to generate uncertainty estimates for the model predictions, addressing the overconfidence issue typically associated with ANNs (Kristiadi et al., 2020). It is important to note that LLLA captures only epistemic uncertainty. Aleatoric uncertainty remains, as predictions are inherently constrained by the data on which they are based. After applying LLLA, the model's adjusted confidence values are shown in Figure 4 (B), where lighter colors indicate higher uncertainty. Some areas, like the western edge and the river region, showed almost no change, but overall, the average confidence dropped from 96.22 % to 88.66 %. This decrease shows that the LLLA method helped adjust the ANN's confidence to be more realistic. When looking at areas where the ANN made correct predictions, the mean confidence decreased by 5.97 %, so just minor adjustments are needed. A larger reduction can be observed in areas where

the predictions are wrong. The mean confidence decreased more, by 9.00 %. This shows that LLLA was effective in reducing the model's overconfidence, especially where it previously made incorrect predictions. Considering spatial differences inside correctly predicted areas, confidence reductions mainly occur along the edges. The opposite is true for wrongly predicted areas, where larger reductions occur more in the center, which is particularly apparent in the northern and southern regions. That indicates, that it is important to look at the spatial variability of the single soil units. A first insight is provided by the violin plot (Figure 5), where the orange-colored part shows the confidence after LLLA. It shows that the spread of confidence values has increased for some soil units, like units 0, 5, 6, 11, and 12. This suggests that LLLA made the model uncertainty more precise for these units. For other units, like 2, 4, 9, and 13, the confidence distribution stayed mostly the same. The highest points of confidence, called peaks, shifted for some units. For example, units 0, 5, and 6 still had high peaks, but units 11 and 12 showed much lower peaks after LLLA. For most other units, the peaks remained high, meaning the ANN stayed confident in its predictions for those units.

Examining the behavior of individual soil units gives further insights. For soil unit 0, which was completely misclassified, the confidence in the areas, where it was predicted, dropped significantly, and the same happened for areas where this soil unit was wrongly predicted. In contrast, soil unit 1, which was mostly misclassified as soil unit 2, maintained high confidence in the correct areas, except for one point in the north, where it was misclassified as soil unit 4. This pattern indicates that when the model predicted soil unit 2, a familiar and similar soil unit, it remained confident, whereas the misclassification to soil unit 4, a less related unit, triggered a higher uncertainty adjustment. This finding indicates that the model is capable of differentiating between plausible misclassifications and more significant errors. A similar pattern is observed with soil unit 3, which was often misclassified as soil unit 4, a closely related soil unit, where the confidence remained high after LLLA. The most extreme example of a plausible misclassification is soil unit 9, which was misclassified as unit 13 for a large area in the west. Notably, there was no reduction in confidence despite the large spatial error, suggesting that the LLLA method failed to detect the misclassification, likely because soil units 9 and 13 are very similar. This highlights a limitation of the LLLA adjustment when soil units have closely overlapping characteristics, making it difficult for the model to recognize the need for uncertainty in such cases. Soil unit 2, which was well-predicted overall, maintained high confidence in both the correct areas and the false positive areas, where it was misclassified as soil unit 1. In the correct areas, it retained the highest confidence levels of all soil units, indicating that the ANN remained highly confident after the LLLA in its accurate predictions. The behavior of the LLLA for soil unit 4 is more complex. Recall that it was overestimated in the center and north and underestimated in the south. After applying LLLA (Figure 4 (B)), the confidence decreased significantly in the central areas, where it was wrongly predicted over soil units 5, 9, and 12, indicating that the model recognized uncertainty in those regions. However, in most of the other misclassified areas, with a small exception in the north, the model's confidence remained high, suggesting that it did not adjust sufficiently for those errors. This lack of uncertainty adjustment could be explained by the large proportion of soil unit 4 in the training data, leading the model to overtrust its predictions for this unit. The model seems to favor overrepresented units, even when faced with evidence of misclassification, which highlights the importance of balancing the training dataset to avoid overconfidence in dominant soil units (Kotzé and van Tol, 2023). The soil units 7 and 8, both absent from the training data, were misclassified as units 4 and 6. However, these misclassifications were detected extremely well by the LLLA adjustment.

Initially, the model assigned high confidence to these areas, but after LLLA, the regions corresponding to soil unit 8 showed some of the lowest confidence values. This indicates that LLLA effectively identified areas of high uncertainty, particularly where the model faced unknown soil units, suggesting that the method is highly effective in detecting errors related to unfamiliar inputs. Soil unit 6 was underestimated overall, with false predictions along the southern margins. In the correct areas, where this unit should have been identified, LLLA significantly lowered the confidence, indicating that the model recognized the initial overconfidence. However, in the wrongly predicted areas, the confidence decrease varied spatially. In the south and east, the model remained highly confident, even where soil unit 6 was misclassified as unit 4, showing that LLLA had difficulties detecting the uncertainty of soil unit 4. This is another example of overtrusted predictions of the model in regions where dominant units like soil unit 4 were prevalent in the training set. Conversely, in the center and north, LLLA effectively detected the misclassification, leading to a clear reduction in confidence. In conclusion, the application of LLLA effectively addressed the overconfidence issue of the ANN by providing uncertainty estimates and adjusting confidence levels in both correct and incorrect predictions. The method successfully reduced confidence in misclassified areas, particularly for unknown soil units like 7 and 8, indicating its effectiveness in detecting unfamiliar inputs. However, the results also show regional variability in the uncertainty adjustments. For some soil units, such as unit 4, the model remained overconfident in dominant units, especially for units that were prevalent in the training data. This highlights a limitation of LLLA in handling closely related or overrepresented soil units, emphasizing the need for balanced training data to improve the model's uncertainty calibration and overall robustness. Nevertheless, compared to broader global approaches like Homosoils, where even the study by (Nenkam et al., 2022) acknowledged that model accuracy improved significantly when incorporating local data, LLLA provides a key advantage by offering spatially resolved uncertainty estimates. This allows for more localized and detailed insights into the reliability of predictions, making it a valuable tool for identifying regional variations in model performance and improving uncertainty calibration at finer scales. In addition to established uncertainty quantification methods for ANNs, such as MC Dropout, ensembles and full Bayesian neural networks, the application of LLLA presents a practical and computationally efficient alternative. As a post hoc method, LLLA enables uncertainty estimation to be incorporated after model training without requiring any modifications to the architecture or learning process. This simplicity made it especially attractive for our soil prediction task, where retraining the model or restructuring the network would have been costly and unnecessary. LLLA operates by approximating the posterior distribution of the final layer weights, capturing model uncertainty with a single forward pass at inference time. Though the computation of the Hessian or its approximation introduces a one-time cost, it does not impact the efficiency of prediction, unlike MC Dropout which multiplies inference costs with repeated forward passes (Daxberger et al., 2021; Kristiadi et al., 2020). In our case, LLLA was highly effective at mitigating overconfidence and highlighting spatial uncertainty in the extrapolation domain, especially in under-sampled areas, confirming its value as a robust, scalable, and lightweight uncertainty quantification tool for DSM applications.

## 4 Conclusion

This study explored the use of Artificial Neural Networks (ANNs) for extrapolation tasks in digital soil mapping (DSM) in under-sampled regions and proposed a novel uncertainty quantification approach using Last-Layer Laplace Approximation (LLLA). The uneven distribution of soil samples limits the reliability of models when extrapolating to new areas. Our research addressed this issue by training an ANN on soil data from a reference area and applying it to a similar but unsampled target area. The results showed that while the ANN could recognize familiar soil patterns, it often produced overconfident predictions, particularly in regions outside the training domain. By applying the LLLA method, we successfully reduced the overconfidence of the ANN and generated spatial uncertainty estimates. This approach provided more realistic confidence values and identified regions where the model's predictions were less reliable. Importantly, LLLA was particularly effective in detecting areas with unfamiliar soil units, reducing confidence in those regions and highlighting the need for further data collection. Our findings underline the importance of uncertainty quantification in DSM, particularly when using machine learning models in spatially diverse landscapes. While ANNs excel in recognizing patterns and extrapolating soil units, their inherent "black-box" nature and tendency for overconfidence pose significant risks when models are deployed in new areas. The LLLA method offers a practical, computationally lightweight solution to address these issues, making it a valuable tool for improving the reliability of soil predictions.

Future work should focus on improving the balance and representativeness of training datasets to enhance the accuracy of uncertainty estimates. Integrating spatial uncertainty maps into sampling strategies can further optimize data collection by directing limited resources to regions of high model uncertainty. Additionally, research should examine how the LLLA responds to established strategies for improving model transferability through the targeted addition of samples, as shown for example by Broeg et al. (2023). In particular, evaluating how LLLA-based uncertainty estimates evolve with such sample augmentation under transfer learning conditions is essential. To further assess the generalizability of LLLA, systematic benchmarking on diverse datasets is necessary. A valuable foundation for this purpose offers for example the LimeSoDa dataset collection, with its broad range of environmental conditions and standardized DSM features (Schmidinger and Heuvelink, 2023). In conclusion, our research demonstrates that combining ANNs with post-hoc Bayesian uncertainty quantification techniques can significantly enhance the interpretability, reliability, and transferability of DSM models. This advancement is essential for making machine learning models more robust and trustworthy in practical applications, particularly in regions with sparse or uneven soil data.

*Code availability.* The code used to perform the analyses and generate the results presented in this study is available upon reasonable request from the corresponding author.

Data availability. The datasets generated and/or analyzed during the current study are not publicly available but are available from the corresponding author upon reasonable request.

Author contributions. KR: Conceptualization, Formal analysis, Methodology, Software, Data Curation, Writing - Original Draft. KE: Software, Validation, Writing - Review & Editing. FS: Software, Formal analysis, Writing - review & editing. MB: Data curation, Writing - review & editing. PH: Supervision, Funding acquisition. TS: Writing - Review & Editing, Supervision, Funding acquisition.

Competing interests. The authors declare that they have no known competing financial interests or personal relationships that could have appeared to influence the work reported in this paper.

Acknowledgements. This research was funded by the Deutsche Forschungsgemeinschaft (DFG, German Research Foundation) under Germany's Excellence Strategy – EXC number 2064/1 – Project number 390727645 and the Tübingen AI Center (FKZ 01IS18039A). We also thank the Department 9: State Authority for Geology, Mineral Resources and Mining (LGRB), Freiburg, Germany, for providing soil data.

The authors acknowledge the use of AI-based tools, specifically OpenAI's ChatGPT and DeepL, to assist with language editing and grammar refinement of this manuscript. The content, interpretations, and conclusions remain entirely the responsibility of the authors.

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
