# Peer review of "Quantifying spatial uncertainty to improve soil predictions in data-sparse regions"

_EGUsphere, 2025_

## Author Comment (AC1)

**Response to reviewer comments on**

**manuscript SOIL egusphere-2025-166:**

**"Quantifying spatial uncertainty to improve soil predictions in data-sparse regions"**

Thanks to the editor and reviewers for the feedback. We've gone through all the comments and made the changes to the manuscript according to your suggestions. This includes fixing specific points that were raised as well as general improvements to the writing, grammar, and overall readability. The revised version should now better reflect the clarity and structure expected. Below, we address the major and the minor comments in detail.

**RC1:**

The use of the uncertainty quantification approach through the Last-Layer Laplace Approximation (LLLA) is a novel and much-needed addition to Digital Soil Mapping (DSM). Artificial Neural Networks (ANNs) are often overconfident, but this approach appears to mitigate that risk. The importance of uncertainty quantification in DSM is increasingly recognized. Nowadays, many people use machine learning algorithms without fully considering the risks of overfitting or overconfidence, which highlights the need for accurate uncertainty measurement, whether in interpolation or extrapolation purposes. Overall, I find the general concept of the paper to be quite interesting.

We thank the reviewer for the positive and supportive feedback. We're pleased that you find the concept of using LLLA for uncertainty quantification in DSM both relevant and promising. Your comments align well with our motivation to address overconfidence in ANN-based soil models. Below, we briefly respond to your suggestions for improvement.

1. However, it could be improved by providing more clarity and adding further details to the methodology section.

   Thank you for the suggestion. While no major methodological changes were required, we acknowledge the need for improved clarity. Combined with the comments of Reviewer 2 on this issue, we have expanded and clarified key parts of the Materials and Methods section.

   Lines 155-156:
   *"A detailed description of the model tuning protocol is provided in Rau et al. (2024), where the method was first tested in a simplified, controlled soil classification setup."*

   Lines 171-172:
   *"To address these limitations and quantify model uncertainty, i.e. epistemic uncertainty, we employ the Last-Layer Laplace Approximation (LLLA) following Daxberger et al. (2021b)"*

   Lines 183-184:
   *"We already have shown that LLLA effectively identifies areas of high uncertainty in soil classification tasks (Rau et al., 2024), making it crucial for generating uncertainty maps with uneven training data coverage"*

2. The results and discussion sections are well written, but the readability would be enhanced if the authors more frequently referenced specific figures. I would recommend this paper for publication in EGU Sphere, pending minor adjustments.

We have reviewed the Results and Discussion sections and improved figure referencing throughout. We now explicitly link specific parts of the narrative to the corresponding figures (particularly Figures 2 to 5) to enhance readability.

Lines 192-193:
*"The study was therefore based on two different areas: a well-sampled reference area and a completely unsampled target area (Figure 1)."*

Lines 258-260:
*"When examining the relationship between confidence and prediction accuracy, it can be seen that in areas where the ANN performs poorly (Figure 2 (B)), the confidence values paradoxically remain high (Figure 4 (A))."*

Lines 290-291:
*"A first insight is provided by the violin plot (Figure 5), where the orange-colored part shows the confidence after LLLA."*

Lines 312-314:
*"After applying LLLA (Figure 4 (B)), the confidence decreased significantly in the central areas, where it was wrongly predicted over soil units 5, 9, and 12, indicating that the model recognized uncertainty in those regions."*

---

## Author Comment (AC2)

**Response to reviewer comments on**

**manuscript SOIL egusphere-2025-166:**

**"Quantifying spatial uncertainty to improve soil predictions in data-sparse regions"**

Thanks to the editor and reviewers for the feedback. We've gone through all the comments and made the changes to the manuscript according to your suggestions. This includes fixing specific points that were raised as well as general improvements to the writing, grammar, and overall readability. The revised version should now better reflect the clarity and structure expected. Below, we address the major and the minor comments in detail.

**RC2:**

The manuscript by Rau and co-authors addresses an important issue for the use of machine learning (ML) models for digital soil mapping, namely the problem of spatial uncertainty. They propose an approach based on a previously published approach combining neural networks (ANN), Bayesian learning and Laplace approximation. The advantage is that this approach informs on spatial uncertainty. The proposed approach is applied to soil classification in central Baden Württemberg in Germany. The manuscript is well organized, and the presentation of the methods and results are clear. Yet, many aspects (listed below) remain unclear and even unjustified. They should be clarified and further elaborated before publication. Therefore, I recommend major corrections by incorporating, if possible, the following recommendations.

We thank the reviewer for the constructive and detailed feedback on our manuscript. We appreciate the recognition of the importance of addressing spatial uncertainty in digital soil mapping and the clear acknowledgment that the manuscript is well organized and clearly presented. We agree that several aspects required further elaboration and clarification, and we have revised the manuscript accordingly. Below, we respond point-by-point to each of the major and minor comments. For each comment, we detail how we have addressed it in the revised version of the manuscript. Where changes were made, we indicate the relevant sections and figures.

**Main comments**

1. Position of the study: In several places in the main text, the authors refer to their previous work published in 2024. It is difficult to see the differences because in that work the authors also address the problem of uncertainty with ANN by combining it with techniques similar to those described in this new study. Could the authors elaborate more on the differences with this study and on the originality of this new work?

    Thank you for highlighting the need to clarify the relationship between this manuscript and our previous work published in 2024 (Rau et al., 2024). That earlier study introduced the use of the Last-Layer Laplace Approximation (LLLA) in the context of digital soil mapping and focused on demonstrating its theoretical potential to improve uncertainty quantification in ANN-based soil models. Additionally, it aimed to systematically explore how ANNs behave, when applied to areas outside their training distribution. The 2024 work was intended as a foundational methodological contribution, relying on simplified and controlled setup to examine the core properties of LLLA and the limitations of standard ANN confidence estimates.

Building on that foundation, the current manuscript moves beyond method development to focus on a practical, spatially explicit application. Specifically, we test how well the ANN with the LLLA approach performs in a real-world soil classification extrapolation task, using a geographically distinct but environmentally similar target region. The two regions were selected based on expert domain knowledge adopted from the German soil survey and a computed similarity index (cosine similarity), providing a meaningful test case for examining spatial generalization, model overconfidence, and the added value of uncertainty quantification.

To clarify these distinctions, we have rewritten all relevant parts of the manuscript where we refer to our previous work (Rau et al., 2024):

Lines 84-92:
*"Such improvements are crucial to ensure that machine learning models for digital soil mapping provide both accurate and reliable predictions. Our previous work Rau et al. (2024) introduced for digital soil mapping the Last-Layer Laplace Approximation (LLLA), a computationally efficient technique that addresses these challenges. Building on this methodological foundation, the current study applies an artificial neural network (ANN) model to an extrapolation task, predicting soil units non-adjacent target area outside the training area. To identify and correct the overconfidence of the ANN and perform a spatial analysis of the model's predictions and associated uncertainties, we use the Last-Layer Laplace Approximation (LLLA), providing corrected uncertainty estimates for every pixel in the target area. Through this, we assess the transferability of the ANN by improving its interpretability and reliability for soil mapping tasks."*

Lines 155-156:

*"A detailed description of the model tuning protocol is provided in Rau et al. (2024), where the method was first tested in a simplified, controlled soil classification setup."*

Lines 182-185:

*"We already have shown that LLLA effectively identifies areas of high uncertainty in soil classification tasks Rau et al. (2024), making it crucial for generating uncertainty maps with uneven training data coverage."*

2. Definition of uncertainty: My second comment may be related to my first one. I am really confused about the type of uncertainty that the authors aim to tackle:
   - The authors underline that the proposed method addresses the problem of spatial uncertainty,
   - At line 155, the authors speak about model parameters and structural uncertainty similarly as the problem of tuning of machine learning models (e.g. Probst et al., 2019). –
   - The title suggests more a problem of data scarcity.
   - The application case with two separate regions seems more related to a problem of transferability (e.g. Ludwig et al., 2023).

References:
Ludwig, Marvin, et al. "Assessing and improving the transferability of current global spatial prediction models." Global Ecology and Biogeography 32.3 (2023): 356-368.

Probst, P., Boulesteix, A. L., & Bischl, B. (2019). Tunability: Importance of hyperparameters of machine learning algorithms. Journal of Machine Learning Research, 20(53), 1-32.

Thank you for this detailed and important question. We acknowledge that the different terms used in the manuscript may have caused confusion. We clarify the types of uncertainty we address and how each aspect connects to specific sections of the manuscript including the references you mentioned.

In general, from the perspective of probability theory, there is no strict separation between data scarcity, structural uncertainty, and transferability: all are manifestations of *epistemic uncertainty*. That is, they reflect the fact that limited data leaves many plausible explanations or parameter configurations open. Mathematically, this corresponds to the geometry of the ANN loss landscape, which can be interpreted as the log posterior in a Bayesian setting. The Laplace approximation provides a local quadratic summary of this geometry and hence captures all forms of epistemic uncertainty. For example, data scarcity in certain regions or classes can lead to flat curvature (i.e., high uncertainty) in the associated parameters, which in turn affects predictive uncertainty in similar but unseen regions, what might appear as a transferability issue. We have revised the manuscript accordingly:

Lines 177-180:

*"Epistemic uncertainty is captured by the LLLA through the local curvature (Hessian) of the loss: flat directions in this geometry indicate parameters that are weakly constrained by the data and thus remain uncertain. Such flatness arises in regions of limited data, structural uncertainty, or poor transferability, all of which reflect ambiguity in the posterior distribution over model parameters."*

• Spatial uncertainty: Yes, the core focus of our study is spatial uncertainty. As described in the Methods section (Lines 191–192) the Last-Layer Laplace Approximation (LLLA) enables pixel-wise probabilistic outputs from the neural network, allowing us to assess how confident the model is for each location in the map.

• Model parameters and structural uncertainty (Line 155, Probst et al., 2019)

We acknowledge the mention of model-related uncertainty may have implied a broader scope than intended. However, as stated in the manuscript (Lines 187–191), we intentionally use a simple, general-purpose ANN architecture rather than optimizing for maximum predictive performance. This decision reflects real-world scenarios where fast deployment of pre-trained models is preferred, and where assessing transferability and uncertainty is more critical than achieving the best possible accuracy. Thus, our primary focus is not on structural uncertainty or tunability, but rather on understanding the model's behaviour in unfamiliar regions through spatial uncertainty.

Lines 189-191:

*"Our goal was to assess and enhance the neural network's ability to extrapolate, and not to achieve the highest possible overall accuracy outperforming other state-of-the-art ANNs, an objective that could be pursued through deliberate and targeted hyperparameter optimization (Probst et al, 2019)."*

• Data scarcity (Title)

We appreciate this observation. While our method does not directly model data scarcity, it is indeed a motivating context for our study. We clarified this point in the manuscript, i.e. Lines

193-195 and Lines 353–360. Specifically, we assume that data-scarce regions are those lacking training data, and our goal is to use spatial uncertainty estimates to identify such regions and inform future data collection strategies.

• Transferability (Ludwig et al., 2023)

You are correct that our design relates to model transferability, but we emphasize that our scope is local rather than global. Unlike Ludwig et al. (2023), who address global spatial prediction models across continents, our study focuses on transferring a model between two environmentally similar regions within the same geographic area. These regions were not arbitrarily chosen they were selected in collaboration with local experts and evaluated using a cosine similarity index, ensuring geoscientific comparability (Lines 133-137).

Lines 137-139:

*"Both the similarity assessment and the expert consultation were carried out in recognition of the fact that, even at the local scale, it is crucial to apply models only where they are valid, a principle already established in global-scale research (Ludwig et al. (2023))."*

Aside from these clarifications, no additional structural changes were made, as the manuscript already reflects these distinctions implicitly. However, we now ensure that readers can more easily distinguish what kind of uncertainty is being addressed, and why.

3. Could the authors clarify the notion of uncertainty that they intend to address? The introduction should be expanded on this aspect, and a discussion of the wide range of uncertainties is also welcome.

To make these points clearer in the manuscript, we now define model uncertainty additionally as epistemic uncertainty in the Methods section (Section 2.4).

Lines 171-172:
*"To address these limitations and quantify model uncertainty, i.e. epistemic uncertainty, we employ the Last-Layer Laplace Approximation (LLLA) following Daxberger et al. (2021)."*

Lines 177-180:
*"Epistemic uncertainty is captured by the LLLA through the local curvature (Hessian) of the loss: flat directions in this geometry indicate parameters that are weakly constrained by the data and thus remain uncertain. Such flatness arises in regions of limited data, structural uncertainty, or poor transferability, all of which reflect ambiguity in the posterior distribution over model parameters."*

In the Discussion, we explicitly state that LLLA does not capture aleatoric uncertainty, as this is related to inherent data noise rather than model knowledge.

Lines 279-281:
*"It is important to note that LLLA captures only epistemic uncertainty. Aleatoric uncertainty remains, as predictions are inherently constrained by the data on which they are based"*

4. Protocol to address spatial uncertainty: If the main objective is to address the problem of spatial uncertainty, I would encourage the authors to carry out more experiments by varying the key factors of the problem: number of training samples, level of similarity between training and test regions, etc. Could the authors propose and carry out a more extensive series of experiments in order to demonstrate the robustness and effectiveness of their approach in a larger number of situations?

We thank the reviewer for this constructive suggestion. We agree that a broader experimental protocol could further strengthen the analysis and we are happy to follow this suggestion in our future work. Our goal in this study was to evaluate how well an ANN with LLLA performs under a realistic local extrapolation scenario. The reference and target regions were selected with expert input and validated using a cosine similarity index to reflect practical DSM applications under data scarcity.
While broader tests (e.g., varying training size or region similarity) are valuable, our access to high-quality, expert-validated data from the State Authority for Geology (LGRB) was limited by logistical and time constraints. We therefore focused on a carefully selected case.
We now include these limitations in the discussion and suggest directions for future work.

Lines 367-374:
*„Future work should focus on improving the balance and representativeness of training datasets to enhance the accuracy of uncertainty estimates. Integrating spatial uncertainty maps into sampling strategies can further optimize data collection by directing limited resources to regions of high model uncertainty. Additionally, research should examine how the Last-Layer Laplace Approximation (LLLA) responds to established strategies for improving model transferability through the targeted addition of samples, as shown for example by Broeg et al. (2023). In particular, evaluating how LLLA-based uncertainty estimates evolve with such sample augmentation under transfer learning conditions is essential. To further assess the generalizability of LLLA, systematic benchmarking on diverse datasets is necessary. A valuable foundation for this purpose offers for example, the LimeSoDa dataset collection, with its broad range of environmental conditions and standardized DSM features (Schmidinger and Heuvelink, 2023). "*

5. Comparison to existing methods: From what I understand of the method proposed by the authors, the ANN is equipped with a final layer for predicting the probability of classification. This is a feature shared by many other techniques, i.e. logistic regression, decision trees, random forest, xgboost, neural networks with Monte Carlo dropout, neural networks combined with a deep set, generative models, and so on. Could the authors elaborate more on the state of the art and discuss the benefits of their method compared to alternative methods?

Thank you for the comment. Our study focuses specifically on methods suitable for ANN models, as the goal is not to compare different machine learning algorithms but to raise awareness of overconfidence and black-box behavior in ANNs. We agree it's important to acknowledge other neural-network-based UQ approaches. In the discussion, we now briefly compare our method to MC dropout, deep ensembles, and Bayesian neural networks, highlighting their trade-offs. We emphasize that LLLA is post-hoc, computationally efficient, and provides pixel-wise uncertainty, making it a practical choice for DSM. We also added a sentence in the Introduction to clarify the broader landscape of UQ in soil science and machine learning.

Line 77-83:

*"The most commonly used methods for uncertainty quantification in DL algorithms, particularly in ANNs, include Monte Carlo (MC) Dropout, ensemble methods, and full Bayesian approaches. These methods, while effective, often require significant computational resources and memory (Abdar et al., 2021). These techniques have begun to gain traction in soil science applications, particularly for estimating uncertainty in soil moisture retrieval or soil spectral models (Li et al., 2023). For example, Padarian et al. (2022) and Huang et al. (2025) utilized these approaches to assess uncertainty in their models, demonstrating their relevance and utility despite the computational demands. These findings underscore the urgent need for methodological advancements that go beyond variance estimation to also tackle overconfidence together with spatial uncertainty while remaining computationally efficient and easy to integrate into existing workflows."*

Line 341-351:

*"In addition to established uncertainty quantification methods for ANNs, such as MC Dropout, ensembles and full Bayesian neural networks, the application of LLLA presents a practical and computationally efficient alternative. As a post hoc method, LLLA enables uncertainty estimation to be incorporated after model training without requiring any modifications to the architecture or learning process. This simplicity made it especially attractive for our soil prediction task, where retraining the model or restructuring the network would have been costly and unnecessary. LLLA operates by approximating the posterior distribution of the final layer weights, capturing model uncertainty with a single forward pass at inference time. Though the computation of the Hessian or its approximation introduces a one-time cost, it does not impact the efficiency of prediction, unlike MC Dropout which multiplies inference costs with repeated forward passes (Daxberger et al., 2021a; Kristiadi et al., 2020). In our case, LLLA was highly effective at mitigating overconfidence and highlighting spatial uncertainty in the extrapolation domain, especially in under-sampled areas, confirming its value as a robust, scalable, and lightweight uncertainty quantification tool for DSM applications."*

Minor comments:

6. Line 55: the authors underline that the ANNs make predictions through complex internal processes that are difficult to understand and interpret. Here references to recent studies improving the interpretability of such methods for digital soil mapping should be added. Suggested references
Padarian, J., McBratney, A. B., and Minasny, B.: Game theory interpretation of digital soil mapping convolutional neural networks, Soil, 6,389–397, 2020.
Wadoux, A. M. J.-C. and Molnar, C.: Beyond prediction: methods for interpreting complex models of soil variation, Geoderma, 422, 115 953, 2022.

Thank you for the suggestion. We have added the recommended references.

Line 55-60:

*"Recent studies have addressed this limitation by introducing model-agnostic interpretation techniques and game theory-based Shapley additive explanations (SHAP), which provide valuable insights into the relationships between environmental covariates and*

*model predictions (Padarian et al., 2020a; Wadoux and Molnar, 2022). In addition, ANNs typically lack built-in uncertainty quantification, which complicates the evaluation of their predictive reliability and may lead to misinterpretations or suboptimal decision-making (Guo et al., 2017)."*

7. The results in Fig. 5 are very convincing. Despite the efficiency of LLMA, a long tail in the probability distribution still remains. I wonder whether this could be further alleviated with an extra calibration of the probability. See for instance Niculescu-Mizil & Caruana (2005). Suggested reference Niculescu-Mizil, A., & Caruana, R. (2005). Predicting good probabilities with supervised learning. In Proceedings of the 22nd international conference on Machine learning (pp. 625-632).

   Thank you for this thoughtful suggestion. We agree that further calibration techniques, such as those proposed by Niculescu-Mizil & Caruana (2005), could help refine the output probabilities. However, in this study, we chose to evaluate the effect of LLLA without additional calibration in order to isolate its native performance. Therefore, we have not made changes in this regard.